# Manufacturing of Metal–Diamond Composites with High-Strength CoCrCu_x_FeNi High-Entropy Alloy Used as a Binder

**DOI:** 10.3390/ma16031285

**Published:** 2023-02-02

**Authors:** Pavel A. Loginov, Alexander D. Fedotov, Samat K. Mukanov, Olga S. Manakova, Alexander A. Zaitsev, Amankeldy S. Akhmetov, Sergey I. Rupasov, Evgeny A. Levashov

**Affiliations:** Department of Powder Metallurgy and Functional Coatings, National University of Science and Technology “MISiS”, Leninsky Prospekt 4, 119049 Moscow, Russia

**Keywords:** high-entropy alloy, powder metallurgy, tensile strength, mechanical properties, diamond, composite

## Abstract

This paper focuses on the study of the structure and mechanical properties of CoCrCu_x_FeNi high-entropy alloys and their adhesion to single diamond crystals. CoCrCu_x_FeNi alloys were manufactured by the powder metallurgy route, specifically via mechanical alloying of elemental powders, followed by hot pressing. The addition of copper led to the formation of a dual-phase FCC + FCC2 structure. The CoCrCu_0.5_FeNi alloy exhibited the highest ultimate tensile strength (1080 MPa). Reductions in the ductility of the CoCrCu_x_FeNi HEAs and the tendency for brittle fracture behavior were observed at high copper concentrations. The equiatomic alloys CoCrFeNi and CoCrCuFeNi demonstrated high adhesion strength to single diamond crystals. The diamond surface at the fracture of the composites having the CoCrFeNi matrix had chromium-rich metal matrix regions, thus indicating that chromium carbide, responsible for adhesion, was formed at the composite–diamond interface. Copper-rich areas were detected on the diamond surface within the composites having the CoCrCuFeNi matrix due to the predominant precipitation of the FCC2 phase at the interfaces or the crack propagation along the FCC/FCC2 interface, resulting in the exposure of the Cu-rich FCC2 phase on the surface.

## 1. Introduction

Developing novel alloys simultaneously characterized by high strength and ductility is one of the topical problems of modern materials science. A great step forward in that direction was taken 20 years ago as there emerged the concept of high-entropy alloys (HEAs), materials consisting of at least five components with concentrations ranging from 5 to 35 at.% [1,2]. The manufacturing of HEAs combines several approaches to the improvement of mechanical properties: solid solution strengthening [3], matrix strengthening by intermetallic nanoparticles during consolidation or heat treatment [4,5], strengthening due to the TWIP and TRIP effects, etc. [6,7]. The range of the potential applications of HEAs is rather wide due to their excellent mechanical properties, as well as resistance to corrosion [8], heat [9] and radiation [10], biocompatibility [11,12], hydrogen storage capacity [13], etc. These features make HEAs stand out among the conventional alloys and also make them highly in demand in the production of metal–diamond composites for abrasive or thermoconductive needs. In recent years, a number of works have been published that reveal the potential of HEAs as a binder for diamond tools, which is associated with a combination of their high strength, ductility and, as a result, high diamond retention strength and better tool performance [14,15,16,17,18]. Among other positive features, these works show the possibility of intermediate carbide layer formation, performing a protective function. As a potential binder material, HEAs compete with the well-known Co-, Fe-Ni-Co-, and Fe-Ni-Cu-Sn-based binders [19,20]. Besides the metal-diamond composites, CoCrFeNi based HEAs are widely used as a matrix for composites, reinforced with SiC particles or fibers [21,22,23,24,25,26], Al_2_O_3_ particles [27,28,29,30], and many others.

Alloys in the Co-Cr-Fe-Ni system are among the best-studied HEAs. In particular, many researchers have demonstrated that it is possible to manufacture materials with high strength and ductility in the CoCrFeMnNi (the Cantor alloy) [31,32,33,34], AlCoCrFeNi [35,36], and CoCrFeNiV systems [37]. Copper is an ideal candidate for being used as one of the components of Fe-Co-Ni-Cr-based HEAs because its physical and chemical characteristics (its atomic radius, melting point, and Young’s modulus) are close to those of Fe, Co, Ni, and Cr. The high thermal conductivity of copper (~400 W/(m K), being four- to fivefold higher than that of chromium or the iron triad metals (80–100 W/(m K)), is its other advantage, which is especially important for metal–diamond composites. Nevertheless, although it is quite promising to use copper as one of the HEA components, there are many fewer studies focusing on CoCrCu_x_FeNi alloys than on their analogs.

A detailed description of the manufacturing technology and the relationship between the microstructure and mechanical properties of CoCrCu_x_FeNi HEAs was provided in refs [38,39,40,41,42,43]. Unlike other HEAs belonging to the CoCrFeNi family, the mechanical properties of copper-containing alloys are studied only upon compression, thus precluding one from drawing an unambiguous conclusion about whether copper is beneficial. The lack of data on the tensile strength of CoCrCu_x_FeNi HEAs in the scientific literature suggests that these alloys are brittle under such testing conditions. The plausible reasons for the poor mechanical properties of CoCrCu_x_FeNi HEAs include the formation of a two-phase dendritic structure in the cast alloys and phase heterogeneity with respect to the chemical composition due to the low solubility of copper in iron, cobalt, and chromium.

A possible way to solve the problem of manufacturing Co-Cr-Cu-Fe-Ni HEAs that would exhibit a combination of excellent mechanical properties is to form the non-cast types of microstructure using alternative technologies for their production (e.g., the methods based on powder metallurgy). It has been demonstrated in refs. [44,45,46] that single-phase CoCrCu_x_FeNi alloys having a nanocrystalline microstructure can be manufactured employing mechanically alloyed (MA) powder mixtures as starting materials and consolidation methods such as hot pressing (HP) or spark plasma sintering (SPS). The advantages of this technology over casting are as follows. Mechanical alloying can be used to produce solid solutions from metal powders due to high-energy mechanical impacts (multiple collisions with grinding bodies in the milling jars) despite the limitations or even the complete absence of mutual solubility [47]. The HP and SPS methods are used to consolidate powder materials at temperatures below the melting point of the main components. Therefore, the nanocrystalline microstructure inherited from the MA powders is preserved.

A number of studies have demonstrated that it is not always possible to obtain a single-phase material in CoCrCu_x_FeNi alloys produced using the MA + HP/SPS technology. At high copper concentrations and consolidation temperatures, the supersaturated FCC solid solution is decomposed, and a secondary phase is precipitated; this phase is also a solid solution with an FCC structure [48], with copper being the predominant component. Shkodich et al. [45] experimentally determined the solubility limit of copper in the CoCrFeNi FCC solid solution with an equiatomic component ratio; it was 9 at. %. There currently are different opinions as to whether the secondary FCC phase is desirable or not [49,50]. Therefore, it would be interesting to study the properties of HEAs in the Co-Cr-Cu-Fe-Ni system with different copper contents, which allows one to produce both single- and dual-phase HEAs. Another important part of this work is the study of the interaction of CoCrCu_x_FeNi HEAs with a single diamond crystal, in particular, the effect of copper on adhesion.

An objective of this study was to investigate the mechanical properties of CoCrCu_x_FeNi HEAs manufactured by MA and HP, as well as the features of deformation and fracturing of metal–diamond composites containing binders based on these alloys.

The effect of the grain size of MA powders on the mechanical properties of HEAs was analyzed. The novelty of the work lies in determining the optimal copper concentration and creating a specific two-phase microstructure, which allows achieving high strength at tensile loads.

## 2. Materials and Methods

Carbonyl iron powder of VK-3 grade (Sintez-CIP LLC, Dzerzhinsk, Russia) with an average particle size of 9 µm and impurity content ≤ 0.3 wt.%, carbonyl nickel powder of PNK-UT3 grade (Kola Mining and Metallurgical Company, Moscow, Russia) with an average particle size of 10 µm and impurity content ≤ 0.06 wt.%, reduced cobalt powder of PK-1U grade (Hanrui Cobalt Ltd., Nanjing, China) with an average particle size of 1.2 µm and impurity content ≤ 0.03%, electrolytic chromium powder of PM-ERKh grade (JSC Polema, Tula, Russia) with an average particle size of 80 µm and impurity content of 0.05 wt.%, and electrolytic copper powder of PMS-1 grade (JSC Uralelectromed, Verkhnyaya Pyshma, Russia) with an average particle size of 24 µm and impurity content of 0.12 wt.% were used as starting materials. Single-crystal diamond powder with a particle size of 40/45 mesh was employed for producing diamond-based composites. Co, Cr, Fe, and Ni powders were mixed in the equiatomic ratio. Copper was added at an amount of 50 and 100% of the equiatomic content.

The powders were mixed in an Activator-2SL planetary ball mill (PBM) with steel grinding media (rotation frequency, 694 min^−1^, centrifugal factor, 90 g; ball-to-powder ratio, 15:1; treatment duration, 5–30 min). In order to prevent oxidation of the green mixture during mixing, the jars were filled with argon (99.998% purity). This treatment of elemental powders yielded composite powders in the form of coarse agglomerated particles. For grinding them, the mixture supplemented with 10 wt.% isopropanol was subjected to additional treatment in the same mode for 3 min. Diamonds were added to the resulting powder mixture using a Turbula laboratory mixer (mixing time, 3 h).

The grain size distribution of the powder mixtures was measured on an Analysette 22 MicroTec plus laser diffraction particle size analyzer (Fritsch, Idar-Oberstein, Germany).

Compact samples (diameter, 50 mm; height, 5 mm) were prepared from the Co-Cr-Cu-Fe-Ni powder mixtures by hot pressing in a graphite mold on a DSP-515 SA setup (Dr. Fritsch, Fellbach, Germany) in a regime typical for metal-diamond composites manufacturing [51,52]: compaction pressure, 35 MPa; time of exposure to the maximum temperature, 3 min. A higher maximum temperature, 1100 °C, was chosen to intensify the shrinkage processes at hot pressing.

Samples for tensile testing (flat samples with a total length of 50 mm and dimensions of the working part being 20 × 5 × 2 mm) were cut out of the hot-pressed preforms by electrical discharge sawing.

The density of the sintered samples was measured by hydrostatic weighing on a GR-202 analytical balance (AND, Tokyo, Japan) with a 10^−4^ g accuracy. The porosity P of the samples was calculated using Formula (1):P = 1 − (ρ_hyd_/ρ)(1)
where ρ_hyd_ is the hydrostatic density, g/cm^3^; and ρ is the density of alloys, g/cm^3^.

The Vickers hardness of the hot-pressed samples was measured on an HVS-50 hardness testing machine at a load of 100 N. The tensile tests were carried out using an Instron 5966 universal testing machine (Instron, Norwood, MA, USA) with a constant displacement rate of 1 mm/min. The Young’s modulus and ultimate tensile strength (UTS) were calculated from the stress-strain curves using the Bluehill software.

X-ray diffraction (XRD) analysis was carried out on an automated DRON 4-07 X-ray diffractometer (LNPO Burevestnik, St. Petersburg, Russia) using monochromatic Co-Kα radiation in the Bragg–Brentano geometry with a step of 0.1°, exposition time of 3 s and range of the test angle of 30–120°. A graphite monochromator was employed for the monochromatization of radiation. The obtained patterns were processed using the software package developed in NUST MISIS [53]; in this case, the phase composition of the samples and the lattice periods were determined with a relative accuracy of 0.0015. Semiquantitative analysis of phase composition, based on the Rietveld refinement method, was carried out using the software package.

The structure of powdered and compacted materials was studied by scanning electron microscopy on an S-3400N microscope (Hitachi, Tokyo, Japan) equipped with a NORAN energy-dispersive X-ray spectrometer (EDX).

## 3. Results

### 3.1. Manufacturing of MA Powder Mixtures

Coarse agglomerated particles were formed from elemental metal powders during the mechanical alloying of Co-Cr-Fe-Ni and Co-Cr-Cu-Fe-Ni powder mixtures. All the used components were ductile, so the structure formation upon MA of powder mixtures took place via the mechanism typical of the “ductile–ductile” systems: upon impact interactions of grinding media, particles were deformed to give rise to new oxide-free surfaces, which formed strong van der Waals bonds with each other [54]. The features of structure formation are discussed in detail for the Co-Cr-Cu-Fe-Ni system (Figure 1). Deformation and agglomeration of initial particles took place during the initial stage of MA. The structure of agglomerates contained easily discernible layers of initial elements. Their thickness, measured by the intercept method, depended on the size of the starting powders and reached 3–5 µm for Fe, Co, and Ni and 20–30 µm for Cr and Cu (Figure 1a). No extensive dissolution of the components was observed, as indicated by the XRD peaks belonging to the corresponding elements (Figure 2, Table 1).

Gradual structure homogenization of composite granules (declining thickness and more chaotic arrangement of the component layers) took place with increasing duration of MA (Figure 1b,c). This stage was characterized by the complete dissolution of the components that had been added as highly dispersed powders (cobalt), as well as by the formation of solid solutions, as indicated by the merging of the XRD peaks belonging to copper and nickel (Figure 2).

Mechanical alloying for 30 min yielded powders with a homogeneous structure (Figure 1). The XRD examination of this powder revealed that it contained only the FCC solid solution having an FCC lattice (lattice parameter a = 0.3609 nm) and a small amount of bcc phase, which is more likely chromium since the chromium powder used is larger than other powders. Therefore, for its complete dissolution, a longer processing time is required.

### 3.2. Grinding of Powders and Determining Their Granulometric Composition

In order to intensify the compaction processes occurring during hot pressing, the MA powder mixtures of each composition were subjected to refinement. For this purpose, they were further treated in the PBM in the presence of 10% isopropyl alcohol. Figure 3a,b shows that this treatment allowed one to significantly reduce the average grain size and obtain a narrow-fraction powder, as exemplified by differential particle size distribution curves of the Co-Cr-Cu-Fe-Ni powder mixtures. The De Brouckere mean diameter D[4,3] of MA Co-Cr-Cu-Fe-Ni powders was 96–194 μm, and the agglomeration tendency increased with higher copper concentrations (Table 2). Treatment of powder mixtures in the presence of alcohol made it possible to reduce the average particle size five- to tenfold due to the adsorption-induced reduction in strength when cracks on the particle surface and other microroughness regions contained liquid.

Below, the compact CoCrCu_x_FeNi samples produced from coarse-grained and fine-grained powders will be denoted “Series 1” and “Series 2”, respectively.

### 3.3. Studying the Mechanical Properties and Structure of HEAs

Compact samples were produced from Series 1 and 2 powder mixtures, and their mechanical properties were studied (Table 3). The two sample series differed significantly in terms of hardness, ultimate tensile strength, and ductility. Porosity was one of the key factors affecting the mechanical properties. According to Equation (1), the porosity of the Series 1 samples was 8.5–9.8%, while the Series 2 samples had a nearly twice lower porosity (3.8–4.9%). This difference can be attributed to the fact that fine-grained powders allow one to intensify shrinkage during sintering. Another advantage related to the use of fine-grained powders is that interparticle pores of smaller diameter and near-spherical shape are formed upon their consolidation, which has a beneficial effect on tensile strength [55].

Both series of samples were characterized by the extremal dependence of hardness on copper content; the maximum hardness was observed at a concentration equal to 0.5 of the equiatomic content. Further increases in copper content reduced hardness because of the structural features of the CoCrCu_x_FeNi alloys.

Unlike in the single-phase CoCrFeNi alloy (Figure 4a,b), the FCC2 secondary phase was formed in CoCrCu_0.5_FeNi and CoCrCuFeNi alloys; identically to FCC, this phase was a solid solution with an FCC crystal lattice. The presence of the FCC2 phase in the XRD patterns of these alloys was confirmed by low-intensity peaks located at smaller 2θ angles with respect to the peaks of the initial FCC phase (Figure 5).

As copper concentration rose to become equiatomic, the content of the FCC2 phase in the alloy increased, which can be seen from the increasing intensity of the peaks of this phase in XRD patterns. The XRD data correlate well with the microstructure of CoCrCu_0.5_FeNi and CoCrCuFeNi alloys. The FCC2 phase is represented as light-grey grains of irregular shape. The amount of FCC2 grains increases at higher Cu concentrations (Figure 4c–f), as it appears from a larger fraction of white-contrast FCC2 phase in the microstructure image (Figure 4e,f).

The chemical composition of the FCC2 phase (86% Cu, 7% Ni, 3% Fe, 3% Co, 1% Cr) was identified by EDX. The fact that copper was the predominant element allows one to put forward a hypothesis that this phase was soft and ductile, so alloy hardness decreased as its content rose.

Figure 6 shows the stress–strain curves of the CoCrCu_x_FeNi alloys manufactured from Series 1 and 2 powders.

The compact Series 1 samples were characterized by extremely low ultimate strength and tensile elongation. The four-component CoCrFeNi alloy had the best combination of properties among these samples: UTS = 830 MPa and ε = 2.7%. Doping with copper significantly reduced the alloy ductility. The CoCrCu_0.5_FeNi and CoCrCuFeNi alloys were fractured at low loads without reaching the plastic deformation stage.

All the Series 2 samples exhibited high tensile strength (930–1080 MPa). The four-component CoCrFeNi alloy was also characterized by high ductility (15%). The CoCrCu_0.5_FeNi alloy had the maximum ultimate tensile strength. As copper concentration was increased to the equiatomic ratio with other components, the strength of the alloy decreased while its ductility slightly increased.

The key factors regulating the behavior of CoCrCu_x_FeNi alloys under tensile loads obviously include residual porosity, solid-solution strengthening of the FCC phase by copper, and the single-phase/two-phase type of the structure.

The elimination of residual porosity by intensifying the shrinkage processes upon hot pressing due to the use of fine-grained MA powders undoubtedly has a significant effect on the mechanical properties of the alloys. Figure 4 demonstrates that the Series 1 and 2 samples differed in terms of pore size and shape. In Series 1 samples, the pore size reached 20 µm; their surface was wedge-shaped, which negatively affected mechanical properties since, in this case, pores acted as stress concentration points and crack propagation channels. Exclusively submicron-sized pores having a predominantly spherical shape (Figure 4a) were observed in Series 2 samples; their volumetric content was ≤5%.

In the CoCrCu_0.5_FeNi and CoCrCuFeNi alloys, the beneficial effect of solid-solution strengthening by copper was cancelled by embrittlement due to the precipitation of the secondary phase in the form of elongated FCC2 phase grains. The copper content in the CoCrCu_0.5_FeNi alloy was close to its solubility limit in the FCC phase, due to which the maximum strength was attained. A small excess amount of copper was spent to form the FCC2 secondary phase. The content of this phase rose with an increasing copper concentration in the CoCrCuFeNi alloy, being a potential reason for the loss of strength.

A fractographic examination was carried out to study the effect of the FCC2 phase on the mechanical properties and behavior of the CoCrCu_x_FeNi alloys under tensile loading.

The fracture surfaces typical of the CoCrFeNi alloy had a homogeneous faceted structure, indicative of the ductile fracture behavior of the alloy, including the FCC phase. The average facet size was 400 nm (Figure 7a).

The fractographic examination of the CoCrCu_0.5_FeNi alloy demonstrated that fracture behavior can be described as the combined mode, where the brittle mode is dominating (Figure 7b). The largest portion of the fracture surface also contained sharp-edged facets, demonstrating that substantial plastic deformation took place in the FCC phase before fracturing. There were also larger areas (sized 5–10 µm) with flat faces along which brittle spalling occurred. The chemical composition of these areas cannot be accurately identified because of the rugged fracture topography. However, the size of spalled spots roughly corresponded to that of the FCC secondary phase precipitates, which allows one to make inferences about their effect on the tendency of Cu-containing alloys for brittle fracture behavior. The absence of roughness, slip bands or slip steps within the spalled spots is indicative of the intercrystallite mechanism of main crack propagation; i.e., it was most likely that the alloy fracturing occurred along the FCC/FCC2 interface.

As copper concentration, the FCC2 phase content, and the specific surface area of the interface increased, the fracture behavior became predominantly brittle. The fracture had a smoother surface; the spalled spots were more than 10 µm in size (Figure 7c). The content of facets in the fracture structure was low, suggesting that the crack propagated along the FCC/FCC2 interface.

### 3.4. Manufacturing of Metal–Diamond Composites and Assessing Adhesion between the Binder and Diamond Single Crystal

Metal–diamond composites were manufactured from powders of equiatomicCoCrFeNi and CoCrCuFeNi HEAs. Only the Series 2 fine-grained powders were used for this purpose as they allow obtaining a metal matrix with the highest mechanical properties. The adhesion between the matrix and diamond single crystal, as well as the effect of copper, were assessed according to the fracture behavior and chemical composition of the areas of the metal matrix adhered to the diamond surface after sample degradation.

In accordance with the conventional views of composite material degradation upon deformation, the main crack is expected to propagate in the least strong structural component or along the weakest interface.

The metal–diamond composites were characterized by high adhesion between the matrix and the single diamond crystal for both HEAs. Figure 8a and Figure 9a indicate that the main crack was propagating not only along the metal–diamond interface (which seems to be the most vulnerable area in the composite) but also in the grain body of some single crystals. Furthermore, the matrix tightly adjoined the diamond single crystal without any gaps (Figure 8b). This fact demonstrates that adhesion strength at the metal–diamond interface formed by van der Waals forces and chemical interactions between the structural components was comparable to the strength of a single diamond crystal.

An EDX study of the fractured diamond surface revealed that chromium was the predominant element in the areas of the adhered matrix (Figure 8b,c). It is known that chromium, among all metals contained in the alloy, has the highest affinity for carbon and a tendency to form carbides. It was shown in [51] that chromium, uniformly distributed in an Fe-Co-Ni-Cr powder mixture, forms an intermediate carbide layer at the metal-diamond interface at sintering. Probably, a similar carbide formation also takes place during the hot pressing of diamond with a high-entropy CoCrCuFeNi matrix.

The fracturing of the composites based on the five-component CoCrCuFeNi HEA occurred via a similar mechanism. The predominantly transcrystallite fracturing of the diamond is indicative of high adhesion strength at the metal–diamond interface (Figure 9a). A more thorough examination of the interface revealed matrix detachment from the diamond (gaps). Their emergence was associated with the low ductility of the CoCrCuFeNi HEA and low residual strain before fracturing (Figure 9b).

Numerous regions with an adhered matrix sized ~10 µm were detected on the surface of non-fractured diamonds. An analysis of the EDX spectra recorded for these regions revealed that they were rich in both chromium and copper (Figure 9c). The presence of copper in the areas adjacent to the diamond was quite unexpected since copper does not possess mutual solubility with carbon, chromium, or chromium carbide [56].

The feature of the trajectory of the main crack upon fracturing of the CoCrCuFeNi dual-phase HEAs is the most plausible reason why copper content was increased in the matrix regions on the diamond surface. Previously, a study focusing on the fracturing of the CoCrCuFeNi alloy has shown that the crack was intercrystallite and propagated along the FCC/FCC2 interface. This fracture mechanism also remained predominant when the alloy contained single diamond crystals. The increased copper content in the areas of the adhered matrix on the diamond was observed when the crack propagated along the FCC/FCC2 interface, thus exposing the FCC2 grains to the fracture surface (Figure 10). Another explanation for the increased copper content on the surface of diamond grains can be related to the features of structural component distribution in the metal–diamond composite. The FCC2 phase was formed due to the fracturing of the supersaturated solid solution FCC during hot pressing, which predominantly occurred at the grain boundaries of the initial composite particles of the CoCrCuFeNi alloy manufactured by MA, as shown in Figure 5. Therefore, the binder–diamond interface was also an area of predominant precipitation of the copper-based phase FCC2, explaining why the copper content on the surface of diamond grains at fractures in the metal–diamond composite was increased.

## 4. Conclusions

CoCrCu_x_FeNi alloy powders with a homogeneous structure and near-100% content of the FCC solid solution of all the components have been manufactured by mechanical alloying.Compact samples of CoCrCu_x_FeNi HEAs have been produced using two series of powders: coarse-grained (D[4,3] = 96–194 µm) and fine-grained (D[4,3] = 13–23 µm). The use of fine-grained powders allows one to reduce the residual porosity of the compact samples manufactured by hot pressing twofold, thus improving the strength and ductility of HEAs.The CoCrCu_0.5_FeNi alloy manufactured from fine-grained powders exhibited the highest strength (1080 MPa). The maximum solid-solution strengthening of the matrix based on the FCC phase was attained at this atomic ratio between components, while the concentration of the undesirable FCC2 secondary copper-based phase (composition: 86% Cu, 7% Ni, 3% Fe, 3% Co, and 1% Cr) was no higher than 5%.A reduction in the ductility of the CoCrCu_x_FeNi HEA was observed as the copper concentration in the alloy increased. A possible reason was the low strength of the boundary between the initial FCC phase and the FCC2 secondary phase.The equiatomic alloys CoCrFeNi and CoCrCuFeNi were characterized by high adhesion strength to single diamond crystals. The diamond surface at the fracture of the composites having the CoCrFeNi matrix had chromium-rich metal matrix regions, thus indicating that chromium carbide was formed at the composite–diamond interface. Copper-rich binder areas were detected on diamonds within the composites having the CoCrCuFeNi matrix. The reasons behind that were as follows: the predominant precipitation of the FCC2 phase at the interfaces, the crack propagating along the FCC/FCC2 interface, and exposure of the Cu-rich FCC2 phase on the surface.

## Figures and Tables

**Figure 1 materials-16-01285-f001:**
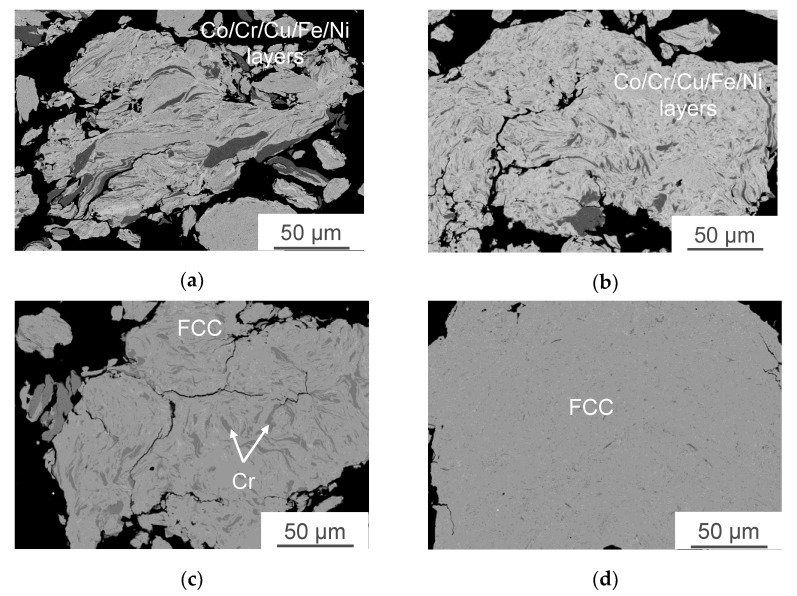
The microstructure of particles of the Co-Cr-Cu-Fe-Ni powder mixtures after MA for 5 (**a**), 10 (**b**), 15 (**c**), and 30 min (**d**).

**Figure 2 materials-16-01285-f002:**
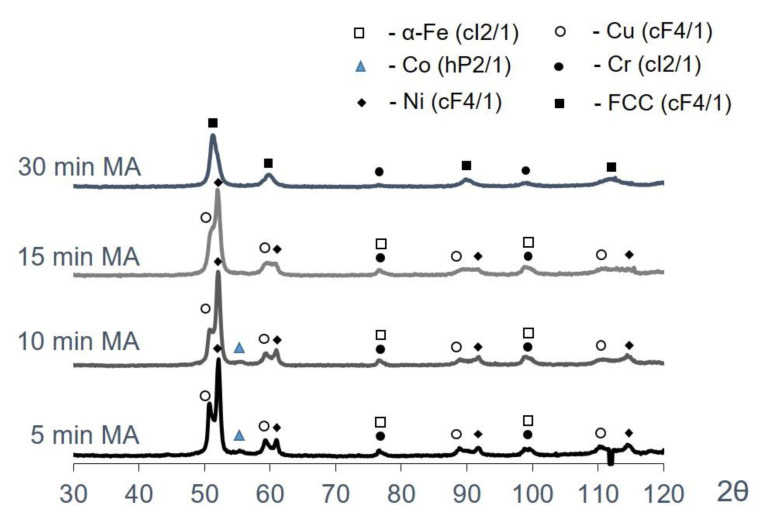
The XRD patterns of the Co-Cr-Cu-Fe-Ni powder mixtures after MA for 5, 10, 15, and 30 min.

**Figure 3 materials-16-01285-f003:**
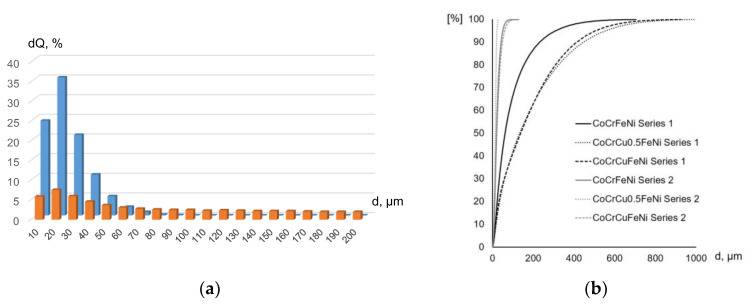
The differential particle size distribution curves of the CoCrCuFeNi powder mixtures before (orange) and after (blue) grinding (**a**) and integral particle size distribution curves of CoCrCu_x_FeNi powder mixtures (**b**).

**Figure 4 materials-16-01285-f004:**
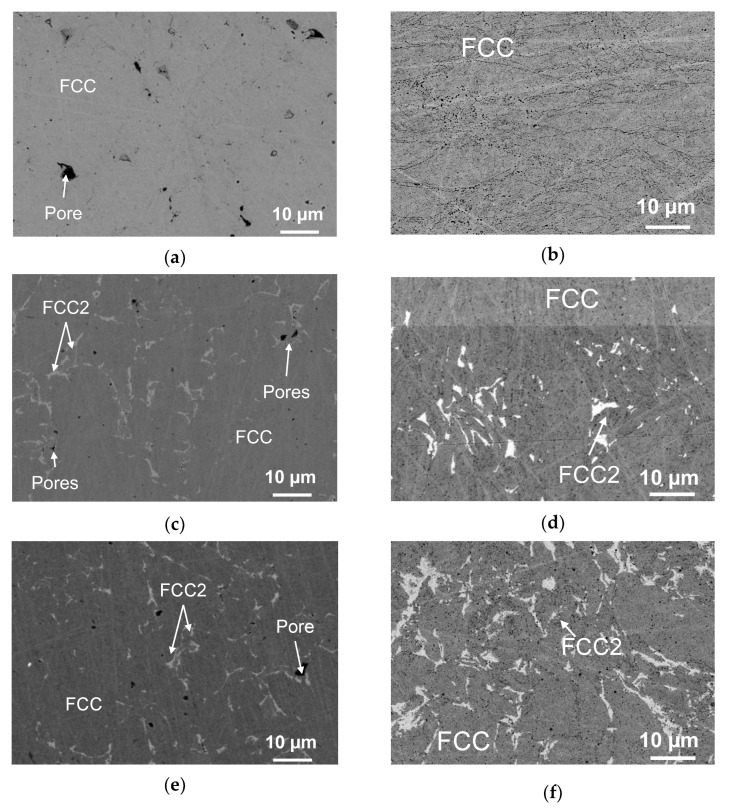
The microstructures of the hot-pressed CoCrFeNi (**a**,**b**), CoCrCu_0.5_FeNi (**c**,**d**), and CoCrCuFeNi (**e**,**f**) samples. Series 1 samples: (**a**,**c**,**e**). Series 2 samples: (**b**,**d**,**f**).

**Figure 5 materials-16-01285-f005:**
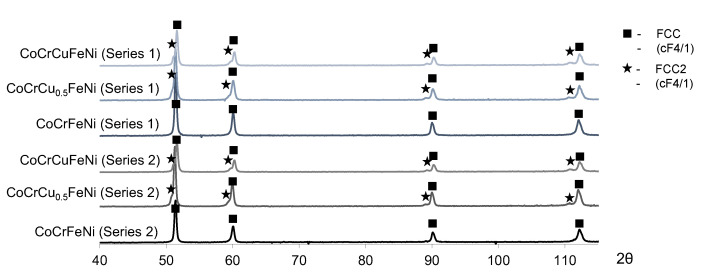
The XRD patterns of the hot-pressed CoCrCu_x_FeNi HEA samples.

**Figure 6 materials-16-01285-f006:**
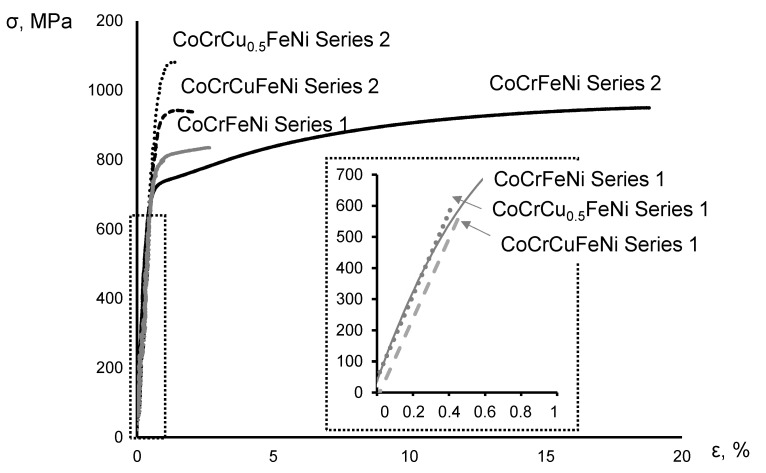
The stress–strain curves for the CoCrCu_x_FeNi alloys manufactured from coarse- and fine-grained powders and the zoomed-in view of the curves for Series 1 samples in the elastic strain range (inset).

**Figure 7 materials-16-01285-f007:**
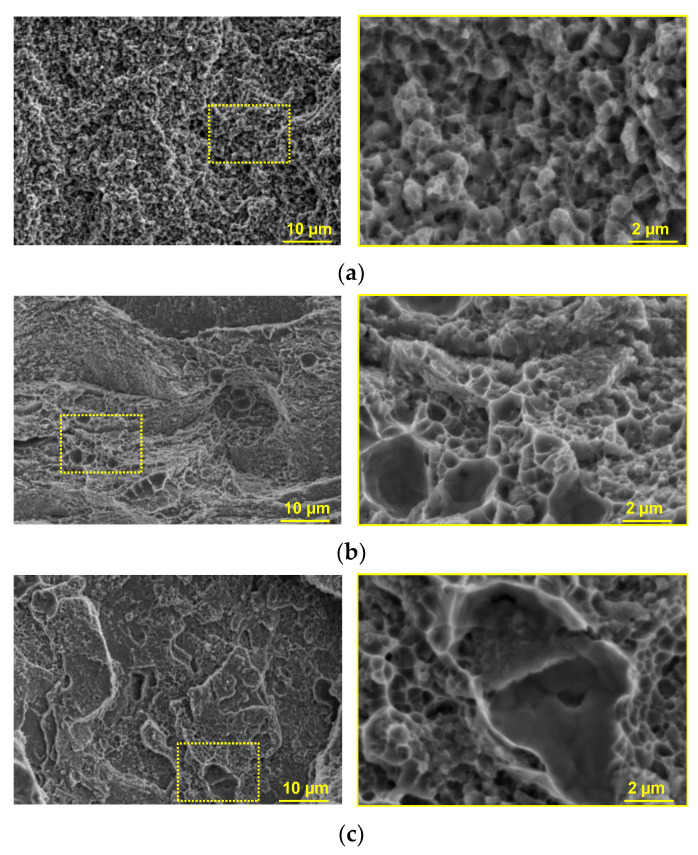
The microstructures of fractures in the CoCrFeNi (**a**), CoCrCu_0.5_FeNi (**b**), and CoCrCuFeNi alloys (**c**). (The areas highlighted by the yellow rectangles show enlarged images of fractures).

**Figure 8 materials-16-01285-f008:**
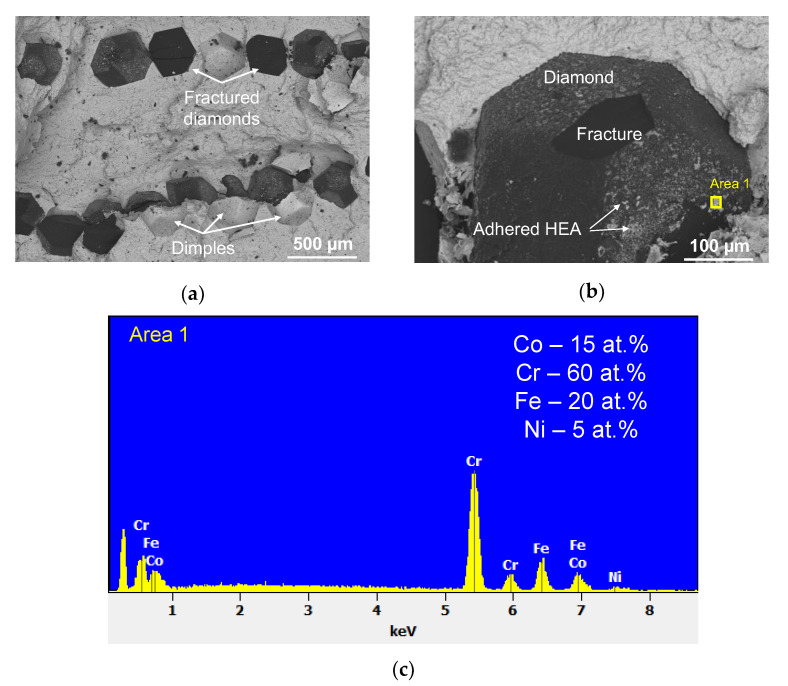
An image of the fracture in the metal–diamond composite with the CoCrFeNi matrix (**a**); an image of the surface of single-crystal diamond (**b**); and the spectrum recorded for the area enclosed by a yellow rectangle (**c**). The chemical composition is shown after subtracting the signal belonging to carbon.

**Figure 9 materials-16-01285-f009:**
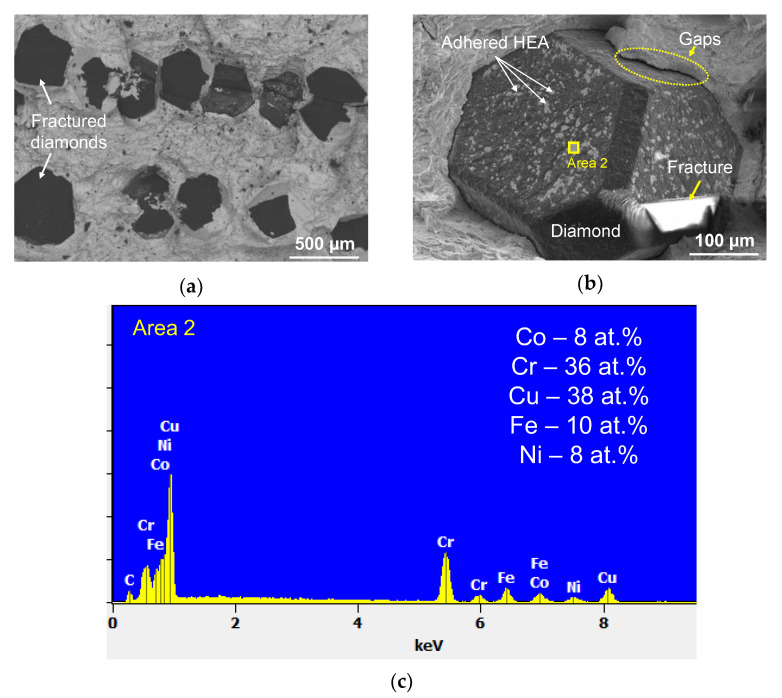
An image of the fracture in the metal–diamond composite with the CoCrCuFeNi matrix (**a**); an image of the surface of a single diamond crystal (**b**), and the spectrum recorded for the area enclosed by a yellow rectangle (**c**). The chemical composition is shown after subtracting the signal belonging to carbon.

**Figure 10 materials-16-01285-f010:**
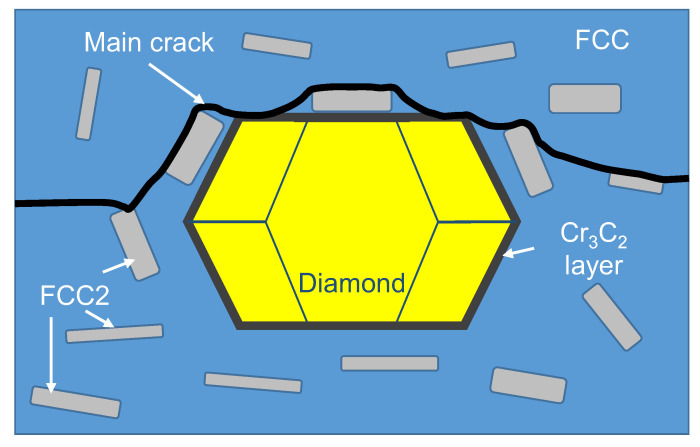
A schematic diagram of the crack trajectory in the metal–diamond composite with CoCrCuFeNi binder.

**Table 1 materials-16-01285-t001:** Phase composition (wt.%) of Co-Cr-Cu-Fe-Ni powder mixtures after MA for 5, 10, 15, and 30 min.

MA Duration	Cr/α-Fe cI2/1	Co hP2/1	Ni cF4/1	Cu cF4/1	FCC cF4/1
5 min	15	7	30	26	22
10 min	12	3	27	20	38
15 min	10	-	21	14	55
30 min	5	-	-	-	95

**Table 2 materials-16-01285-t002:** The particle size distribution of the CoCrCu_x_FeNi powder mixtures.

Composition	D50, µm	D90, µm	D(4,3), µm
Series 1 (coarse-grained mixture)			
CoCrFeNi	61.0	231.8	96.64
CoCrCu_0.5_FeNi	133.0	448.3	189.01
CoCrCuFeNi	151.7	437.2	194.53
Series 2 (fine-grained mixture)			
CoCrFeNi	18.9	41.5	22.04
CoCrCu_0.5_FeNi	13.7	22.8	13.91
CoCrCuFeNi	16.5	39.1	23.28

**Table 3 materials-16-01285-t003:** The mechanical properties of hot-pressed CoCrCu_x_FeNi HEA samples produced from powder mixtures of different grain sizes.

Composition	Porosity, %	Hardness, HV	UTS, MPa	ε, %
Series 1				
CoCrFeNi	9.4 ± 0.5	188 ± 4	830 ± 10	2.7 ± 0.2
CoCrCu_0.5_FeNi	9.8 ± 0.4	290 ± 5	400 ± 10	0.35 ± 0.08
CoCrCuFeNi	8.5 ± 0.4	262 ± 4	580 ± 10	0.35 ± 0.06
Series 2				
CoCrFeNi	3.9 ± 0.5	282 ± 4	930 ± 20	15.0 ± 0.8
CoCrCu_0.5_FeNi	4.9 ± 0.5	372 ± 5	1080 ± 10	1.3 ± 0.1
CoCrCuFeNi	3.8 ± 0.3	327 ± 5	940 ± 10	2.2 ± 0.1

## Data Availability

Not applicable.

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
