# Peer review of "Manufacturing of Metal–Diamond Composites with High-Strength CoCrCuxFeNi High-Entropy Alloy Used as a Binder"

_materials, 2023, doi:10.3390/ma16031285_

Round 1

Reviewer 1 Report

1 The authors should give detail descriptions on the processes to obtain the mechanical properties such as stress-strain curves, measurement of porosity, etc.

2 The authors should provide detail grain size fractions and distributions.

3 Please clarify the conclusion on "ductile-to-brittle transition of 353
fracture behavior were observed as copper concentration in the alloy increased".  It should be noted that ductile-brittle transition needs careful experimetal test based on fracture mechanics approch or strength measurement.

Author Response

We would like to express our thanks to the reviewer for a careful study of our paper and useful recommendations. Obviously, they will improve the quality of our paper. Please find below our point-by-point responses to the comments. All changes in manuscript text were embedded and highlighted in yellow. Revised text contains corrections, made upon reviewer’s recommendations.

Comment 1

The authors should give detail descriptions on the processes to obtain the mechanical properties such as stress-strain curves, measurement of porosity, etc.

Response

We made a number of corrections in Materials and Methods section to give more detailed description of the processes (mechanical alloying, XRD etc.). Among other things, we added, that tensile tests were carried out with constant displacement rate 1 mm/min.

Comment 2

The authors should provide detail grain size fractions and distributions.

Response

We added D50 and D90 parameters in the Table 1 to make the demonstration of particle size distribution more detailed.

The differential particle size distribution was shown only for CoCrCuFeNi powder mixtures to make Figure 3a more readable. But we complemented it with integral particle size distribution for all produced powder mixtures (Figure 3 b).

Comment 3

Please clarify the conclusion on "ductile-to-brittle transition of 353
fracture behavior were observed as copper concentration in the alloy increased".  It should be noted that ductile-brittle transition needs careful experimetal test based on fracture mechanics approch or strength measurement.

Response

Thank you for your comment. We agree, that the used term «ductile-to-brittle transition» is incorrect. We made changes within the text, indicating the tendency of Cu-rich alloys for brittle fracture only.

The corrections were made in Abstract, Section 3.3 and Conclusions.

Reduction of ductility of the CoCrCuxFeNi HEAs and tendency for brittle fracture behavior were observed at high copper concentration.

Reviewer 2 Report

Dear authors,

Thank you for an interesting and topical manuscript. From my point of view, the manuscript is well-structured and clearly written. I found some typos/notes and I have some recommendations for correction:

             Please unify in the whole text: CoCrCuxFeNi vs. CoCrCuxFeNi and FCC1 vs. FCC.

             Line 9: high-entropy

             Line 123: 10-4 g. It should be in a superscript.

             Line 131: Add the company and country of the diffractometer developer. Which software did you use for phase analysis evaluation?

             Line 189: Add “according to eq. (1)”

             Lines 236 and 241: Shouldn’t be there Fig. 4?

             Fig. 6: Keep the same marking of Series.

             Fig. 10: Add the information that the CoCrCuFeNi binder is FCC2 phase.

             In the References, there are no spaces before years and many missing spaces after commas and semicolons.

I have six questions/notes that could be answered:

1.            In the Introduction, there is nothing about metal-diamond composite. It will be suitable to add it.

2.            In the text (related to Fig. 1 and 2), you wrote that after MA for 30 min, there is a small amount of non-dissolved chromium. Which method did you use for Cr identification? Please add the information. You determined the Cr phase in Fig. 1c, but there are 2 bcc phases, Cr and alpha-Fe, in Fig. 2 (15 min). Could you explain it or correct it?

3.            Which crystallographic symmetry (Bravais lattice) belongs to particular phases? Add to Fig. 2 or a new table. The information about wt.% phase content will be helpful too (same after HP). How do you explain a small amount of hcp-Co phase on the XRD pattern?

4.            Explain the reason, why do you use only Series 2 powders for metal-diamond composites (line 283).

5.            Referring to reference 37, you mention that there are Cr3C2 particles on the diamond surface. It would also be appropriate to state the proof, e.g. to use micro-XRD with a narrowly focused beam. For Fig. 9, it is much more important.

6.            In Conclusion 3, you mentioned that there are only 5% (wt. or vol.?) of the FCC2 phase. But you didn’t mention any information about Rietveld refinement (or other methods) for the ratio calculation.

Author Response

We would like to express our thanks to the reviewer for a careful study of our paper and useful recommendations. Obviously, they will improve the quality of our paper. Please find below our point-by-point responses to the comments. All changes in manuscript text were embedded and highlighted in yellow. Revised text contains corrections, made upon reviewer’s recommendations.

Typos/Notes

  • Please unify in the whole text: CoCrCuxFeNi vs. CoCrCuxFeNi and FCC1 vs. FCC.
  • Line 9: high-entropy
  • Line 123: 10-4 g. It should be in a superscript.
  • Line 131: Add the company and country of the diffractometer developer. Which software did you use for phase analysis evaluation?
  • Line 189: Add “according to eq. (1)”
  • Lines 236 and 241: Shouldn’t be there Fig. 4?
  • Fig. 6: Keep the same marking of Series.
  • Fig. 10: Add the information that the CoCrCuFeNi binder is FCC2 phase.
  • In the References, there are no spaces before years and many missing spaces after commas and semicolons.

Response

Thank you. All the typos and small mistakes were corrected. Figure 6 contains the same marking of Series.

Only Figure 10 is left unchanged, because CoCrCuFeNi binder contains both FCC and FCC2 phase.

For phase analysis evaluation we used the software package, developed in NUST MISIS. The detailed description of the software can be found in [E. V. Shelekhov and T. A. Sviridova. Programs for x-ray analysis of polycrystals. Metal Science and Heat Treatmeте 2000. 42(7) 309-313]

Comment 1

In the Introduction, there is nothing about metal-diamond composite. It will be suitable to add it.

Response

We added information about previous works on the application of HEAs as binders for metal-diamond composites in the beginning of the Introduction. And also we mentioned, that the important part of our work is the study of the interaction of CoCrCuxFeNi HEAs with a diamond single crystal, in particular the effect of copper on adhesion.

In recent years, a number of works have been published that reveal the potential of HEAs as a binder for diamond tools, which is associated with a combination of their high strength, ductility and, as a result, high diamond retention strength and better tool performance [14-18]. Among other positive features, these works show the possibility of intermediate carbide layers formation, performing a protective function. As a potential binder material HEAs compete with well known Co, Fe-Ni-Co, Fe-Ni-Cu-Sn based binders [19,20]. Besides the metal-diamond composites CoCrFeNi based HEAs are widely used as a matrix for composites, reinforced with SiC particles or fibers [21-26], Al2O3 particles [27-30] and many others.

Another important part of this work is the study of the interaction of CoCrCuxFeNi HEAs with a diamond single crystal, in particular the effect of copper on adhesion.

Comment 2

In the text (related to Fig. 1 and 2), you wrote that after MA for 30 min, there is a small amount of non-dissolved chromium. Which method did you use for Cr identification? Please add the information. You determined the Cr phase in Fig. 1c, but there are 2 bcc phases, Cr and alpha-Fe, in Fig. 2 (15 min). Could you explain it or correct it?

Response

Our hypothesis, that bcc peaks on XRD patterns are from chromium, is based upon 2 reasons. First, we have studied the polished samples of powders by SEM (in BSE mode) and XRD. As it is shown in Figure 1 c,d, there are some traces of dark grey phase within the uniform matrix. Chromium appears as the darkest regions among all used elements (in BSE). Also it was later confirmed by EDX analysis in the dark areas. They were enriched by Cr. The EDX spectra were not added in the paper for space considerations.

Second, the chromium powder, used in this work, was much coarser, than the iron powder (80 µm vs. 9 µm). So, we suppose, that complete dissolution of chromium in FCC solid solution requires more energy and longer duration of processing.

We left the following explanation in the text of the article.

The XRD examination of this powder revealed that it contained only the FCC solid solution having an fcc lattice (lattice parameter a = 0.3609 nm) and a small amount of bcc phase, which is more likely chromium, since the chromium powder used is larger than other powders. Therefore, for its complete dissolution, a longer processing time is required.

Comment 3

Which crystallographic symmetry (Bravais lattice) belongs to particular phases? Add to Fig. 2 or a new table. The information about wt.% phase content will be helpful too (same after HP). How do you explain a small amount of hcp-Co phase on the XRD pattern?

Response

The types of Bravais lattices were added in Figure 2. Also Table 1, demonstrating phase content, was added

Small amount of hcp-Co phase after 5 minutes of MA could be explained by high velocity of Co dissolution due to very fine particle size of Co powder used (1.2 µm).

Comment 4

Explain the reason, why do you use only Series 2 powders for metal-diamond composites (line 283).

Response

We used only the Series 2 fine-grained powders because they allow obtaining a metal matrix with the highest mechanical properties. It is beneficial for metal-diamond composites, because diamond adhesion strength usually correlates with the strength of metal matrix itself.

The corrections are embedded within the text.

Comment 5

Referring to reference 37, you mention that there are Cr3C2 particles on the diamond surface. It would also be appropriate to state the proof, e.g. to use micro-XRD with a narrowly focused beam. For Fig. 9, it is much more important.

Response

Unfortunately micro-XRD has not been done due to time restrictions. We agree, that in the current form our statement looks too speculative. So it was reformulated. Though, it is worth to be mentioned, that high Cr concentration in adhered matrix regions is very similar to what we have found in our previous work [37] at investigation of metal-diamond lamella by means of TEM. In that work Fe-Co-Ni-Cr powder mixture was also prepared by mechanical alloying technique, Cr was distributed uniformly in the powder. And exactly Cr formed a thin carbide interlayer at the interface.

It is known that chromium among all metals, contained in the alloy, has the highest affin-ity for carbon and a tendency to form carbides. It was shown in [37] that chromium, uni-formly distributed in an Fe-Co-Ni-Cr powder mixture, forms an intermediate carbide layer at the metal-diamond interface at sintering. Probably, a similar carbide formation also takes place at hot pressing of diamond with high-entropy CoCrCuFeNi matrix.

Comment 6

In Conclusion 3, you mentioned that there are only 5% (wt. or vol.?) of the FCC2 phase. But you didn’t mention any information about Rietveld refinement (or other methods) for the ratio calculation.

Response

The phase ratio calculation was carried out using the above mentioned software package, developed in NUST MISIS. It was based on Rietveld refinement method.

The information was added in the Materials and Methods section.

Reviewer 3 Report

The manuscript titled “Manufacturing of metal–diamond composites with high-strength CoCrCuxFeNi high-entropy alloy used as a binder” was reviewed carefully. Author has done good works. The manuscript can be accepted with major corrections given below:

[1] Lot of articles related to use of high entropy alloy as the binder materials for composites, must include their literature survey in introduction.

[2] In Methods and Materials section: Experimental data like Mechanical alloying related (kind of grinding media used, ball to powder ratio were not given);

[3] XRD experimental details like measurement range and step size were not included.

[4] Consider to change the term additive density;

[5] What was the sample specifications, strain rate used in tensile test;

[6] Fraction of diamond used in composite preparation;

[7] Did bend test results were used in this study, if not remove about them and also ultimate tensile strength was mentioned as σb (in table and discussion part) it must be cross checked;

[8] Results: How the layer thickness of each elements layer in powder were measured and how they were distinguished?

[9] Correlations between different results like XRD, microstructures were not made clearly.

[10] XRD related plots, Labels in X and Y axis must be below and left side of the axis line for clear view; XRD of fine powders series, CoCrFeNi, CoCrCu0.5Ni alloys powder milling, compacted series 1 alloys were not given. It is needed to be studied to correlate the results.

[11] Line 209, statement “large quantity of grains of phase” can be written as larger fraction of white contrast FCC phase”.

[12] Scale bars are missing in Fig 4.

[13] Lines 236, 241; Figure 5 must be changed to Figure 4.

[14] SEM microstructures, Performance test results of composite were not presented only fractographs were given and discussed (will not full fill the study as given in the title).

Author Response

We would like to express our thanks to the reviewer for a careful study of our paper and useful recommendations. Obviously, they will improve the quality of our paper. Please find below our point-by-point responses to the comments. All changes in manuscript text were embedded and highlighted in yellow. Revised text contains corrections, made upon reviewer’s recommendations.

Comment 1

Lot of articles related to use of high entropy alloy as the binder materials for composites, must include their literature survey in introduction.

Response

We added the first paragraph of the Introduction with references to the works on HEA based binders for the diamond tools and application of HEAs as matrix for composite materials.

In recent years, a number of works have been published that reveal the potential of HEAs as a binder for diamond tools, which is associated with a combination of their high strength, ductility and, as a result, high diamond retention strength and better tool per-formance [14-18]. Among other positive features, these works show the possibility of in-termediate carbide layers formation, performing a protective function. As a potential binder material HEAs compete with well known Co, Fe-Ni-Co, Fe-Ni-Cu-Sn based binders [19,20]. Besides the metal-diamond composites CoCrFeNi based HEAs are widely used as a matrix for composites, reinforced with SiC particles or fibers [21-26], Al2O3 particles [27-30] and many others.

Comment 2

In Methods and Materials section: Experimental data like Mechanical alloying related (kind of grinding media used, ball to powder ratio were not given)

Response

The description of mechanical alloying methods was supplemented with information about kind of grinding media used, ball to powder ratio.

The powders were mixed in an Activator-2SL planetary ball mill (PBM) with steel grinding media (rotation frequency, 694 min-1, centrifugal factor, 90g; ball-to-powder ratio, 15:1; treatment duration, 5–30 min).

Comment 3

XRD experimental details like measurement range and step size were not included.

Response

XRD experimental details were added in the text.

X-ray diffraction (XRD) analysis was carried out on an automated DRON 4-07 X-ray diffractometer (LNPO Burevestnik, Russia) using monochromatic Co-Kα radiation in the Bragg–Brentano geometry with a step of 0.1°, exposition time 3 s and the range of the test angle is 30–120°.

Comment 4

Consider to change the term additive density

Response

Thank you for your comment. We used a wrong term.

It was replaced with “density”.

Comment 5

What was the sample specifications, strain rate used in tensile test.

Response

Flat samples with a total length of 50 mm and dimensions of the working part being 20×5×2 mm were used for tensile tests. The tensile tests were carried out with constant displacement rate 1 mm/min.

The corresponding changes were embedded in the text.

Comment 6

Fraction of diamond used in composite preparation.

Response

We have used single crystal diamond powder with particle size of 40/45 mesh (~354-400 μm) for producing diamond-based composites.

Comment 7

Did bend test results were used in this study, if not remove about them and also ultimate tensile strength was mentioned as σb (in table and discussion part) it must be cross checked.

Response

The information about sample geometry for bending tests was removed from the text. The ultimate tensile strength was labeled in a traditional way – as UTS.

Comment 8

Results: How the layer thickness of each elements layer in powder were measured and how they were distinguished?

Response

We estimated the layer thickness for MA powders being processed for 5-10 minutes (when the layers are easily distinguished) by intercept method. To distinguish different elements in powder particle cross section we complemented SEM imaging with XRD mapping or spectra in local areas. For space consideration we don’t include XRD data in the manuscript.

Some minor changes were added within the following sentence.

Their thickness, measured by intercept method, depended on size of the starting powders and reached 3–5 µm for Fe, Co, and Ni and 20–30 µm for Cr and Cu (Figure 1a).

Comment 9

Correlations between different results like XRD, microstructures were not made clearly

Response

We added the description to Figure 4 to make the correlation between phase composition and microstructure more clear.

As copper concentration rose to become equiatomic, the content of the FCC2 phase in the alloy increased, which can be seen from the increasing intensity of the peaks of this phase in XRD patterns. The XRD data correlate well with the microstructure of CoCrCu0.5FeNi and CoCrCuFeNi alloys. FCC2 phase is represented as light-grey grains of irregular shape. The amount of FCC2 grains increases at higher Cu concentrations (Figure 4 c-f).

Comment 10

XRD related plots, Labels in X and Y axis must be below and left side of the axis line for clear view; XRD of fine powders series, CoCrFeNi, CoCrCu0.5Ni alloys powder milling, compacted series 1 alloys were not given. It is needed to be studied to correlate the results

Response

Figure 5 was replaced with the new one, containing patterns of samples, prepared from Series 1 mixtures.

Regarding Figure 2, we suppose, that XRD patterns of refined powders would be excessive. It is worth pointing out, that all phase transformations at mechanical alloying can take place at the stage of «dry» milling. Additional milling with isopropanol was carried out only for 3 minutes, and its aim was to unwedge the MA powder particles at phase boundaries, thus preventing mutual dissolution of the elements. Isopropanol here acts as process control agent, which reduces the effect of cold welding. More detailed description of the process can be found in [C. Suryanarayana. Mechanical alloying and milling. Progress in Materials Science 46 (2001) 1-184], P. 26-29.

Comment 11

Line 209, statement “large quantity of grains of phase” can be written as larger fraction of white contrast FCC phase”.

Response

Thank you for your recommendation. Actually, it is a better phrasing.

The sentence was changed.

Comment 12

Scale bars are missing in Fig 4.

Response

Figures 4 c, e were replaced with ones with scale bars.

Comment 13

Lines 236, 241; Figure 5 must be changed to Figure 4.

Response

The corrections were made.

Comment 14

SEM microstructures, Performance test results of composite were not presented only fractographs were given and discussed (will not full fill the study as given in the title).

Response

In this work we made only estimation of metal matrix adhesion to diamond and investigated the chemical composition of the metal on the diamond surface.

SEM microstructures of the composites were not made due to complexity of preparing the diamond-containing polished sample. Performance tests will be included in our future works, when the pilot models of cutting tools, based on high entropy CoCrCuFeNi alloys, are produced.

Reviewer 4 Report

1)    In the ABSTRACT, the sentence “A possible reason was the low strength of the boundary between the initial FCC phase and the FCC2 secondary phase” do not sound properly. Probably this sentence to be removed. Or else to be modified.

2)    Please add the novelty of the work at the end of Introduction section.

3)    The selection of compact sample preparation parameters (page 3) are based upon literature? Please mention and add the literature proof.

4)    Under section 3.3, it is explained as “This difference can be attributed to the fact that fine-grained powders allow one to intensify shrinkage during sintering. Another advantage related to the use of fine-grained powders is that interparticle pores of smaller diameter and near-spherical shape are formed upon their consolidation, which has a beneficial effect on mechanical properties”. It is true. The mechanical properties mean, there is a set of.  Some of the properties like, tensile strength and hardness go hand in hand, whereas toughness and plasticity are inversely varying. Hence, please give the literature support and mention the specific mechanical property interested.

5)    The fracture surface in CoCrCu0.5FeNi alloy (Figure 7b) indicates fracture is the combination of brittle and ductile mode. Please mention the failure mode as the combined mode where brittle mode is dominating.

 In REFERENCE (2, 16, 26, 28, 32, 34 and 38) the format followed for the title of paper is different (1st letter of all words is capital) compared to the remaining. Please correct for the format uniformity

Author Response

We would like to express our thanks to the reviewer for a careful study of our paper and useful recommendations. Obviously, they will improve the quality of our paper. Please find below our point-by-point responses to the comments. All changes in manuscript text were embedded and highlighted in yellow. Revised text contains corrections, made upon reviewer’s recommendations.

Comment 1

In the ABSTRACT, the sentence “A possible reason was the low strength of the boundary between the initial FCC phase and the FCC2 secondary phase” do not sound properly. Probably this sentence to be removed. Or else to be modified.

Response

Thank you for your comment. That sentence was removed from the abstract.

Comment 2

Please add the novelty of the work at the end of Introduction section.

Response

The last paragraph of the introduction was restated, highlighting the novelty of the work. The corrections were embedded in the text.

The novelty of the work lies in determining the optimal copper concentration and creating a specific two-phase microstructure, which allows to achieve high strength at tensile loads.

Comment 3

The selection of compact sample preparation parameters (page 3) are based upon literature? Please mention and add the literature proof.

Response

We added two references, showing the similar hot pressing regimes for manufacturing of metal-diamond composites with Fe-Co-Ni based binders. A higher maximum temperature, 1100°C, was chosen to intensify the shrinkage processes at hot pressing. And also it lies beneath the temperature of appearance of liquid phase, based on Cu, as it was found in supporting experiments by DSC method (~1130 °C). We decided not to show the DSC results in the paper and restated the paragraph, relating to compact sample preparation.

Compact samples (diameter, 50 mm; height, 5 mm) were prepared from the Co-Cr-Cu-Fe-Ni powder mixtures by hot pressing in a graphite mold on a DSP-515 SA setup (Dr. Fritsch, Germany) in the regime, typical for metal-diamond composites manu-facturing [XXX,XXXX]: compaction pressure, 35 MPa; time of exposure to the maximum temperature, 3 min. A higher maximum temperature, 1100°C, was chosen to intensify the shrinkage processes at hot pressing.

Comment 4

Under section 3.3, it is explained as “This difference can be attributed to the fact that fine-grained powders allow one to intensify shrinkage during sintering. Another advantage related to the use of fine-grained powders is that interparticle pores of smaller diameter and near-spherical shape are formed upon their consolidation, which has a beneficial effect on mechanical properties”. It is true. The mechanical properties mean, there is a set of.  Some of the properties like, tensile strength and hardness go hand in hand, whereas toughness and plasticity are inversely varying. Hence, please give the literature support and mention the specific mechanical property interested.

Response

We reformulated the sentence to specify the targeted property – tensile strength. And provided references for a detailed review, describing the effect of the residual porosity of powder metallurgy materials on strength.

This difference can be attributed to the fact that fine-grained powders allow one to inten-sify shrinkage during sintering. Another advantage related to the use of fine-grained powders is that interparticle pores of smaller diameter and near-spherical shape are formed upon their consolidation, which has a beneficial effect on tensile strength [Ternero, F.; Rosa, L.G.; Urban, P.; Montes, J.M.; Cuevas, F.G. Influence of the Total Porosity on the Properties of Sintered Materials—A Review. Metals 2021, 11, 730.].

Comment 5

The fracture surface in CoCrCu0.5FeNi alloy (Figure 7b) indicates fracture is the combination of brittle and ductile mode. Please mention the failure mode as the combined mode where brittle mode is dominating.

Response

Thank you for your comment. The sentence was reformulated.

Fractographic examination of the CoCrCu0.5FeNi alloy demonstrated that fracture behavior can be described as the combined mode, where brittle mode is dominating (Figure 7 b).

Comment 6

In REFERENCE (2, 16, 26, 28, 32, 34 and 38) the format followed for the title of paper is different (1st letter of all words is capital) compared to the remaining. Please correct for the format uniformity

Response

We have checked the formatting of the references and chose lowercase letters for the article titles.

Round 2

Reviewer 1 Report

No further comments